# Interaction effects in the association between methadone maintenance therapy and experiences of racial discrimination in U.S. healthcare settings

**George Pro** [1] *, **Nick Zaller** [2]

**1** Center for Health Equity Research, Northern Arizona University, Flagstaff, Arizona, United States of America, **2** University of Arkansas for Medical Sciences College of Public Health, Little Rock, Arkansas, United States of America

* george.pro@nau.edu

## Abstract

### Background

Disparities in methadone maintenance therapy (MMT) outcomes have received limited attention, but there are important negative outcomes associated with MMT that warrant investigation. Racial discrimination is common in healthcare settings and affects opioid use disorder (OUD) treatment and comorbidities. However, race/ethnicity alone may not fully explain experiences of discrimination. MMT remains highly stigmatized and may compound the effect of race/ethnicity on discrimination in healthcare settings. We sought to quantify differential associations between MMT and experiences of racial discrimination between racial/ethnic groups in a U.S. national sample.

### Methods

We used the National Epidemiologic Survey on Alcohol and Related Conditions-III (2012–2013) to identify a subset of individuals with a lifetime OUD who had ever used MMT (survey n = 766; weighted population n = 5,276,507). We used multivariable logistic regression to model past-year experience of racial discrimination in a healthcare setting. We included an interaction term between race/ethnicity and MMT status to identify the odds of discrimination (MMT vs. no MMT [referent]) within racial/ethnic groups. We used survey procedures with weights to account for the parent study's complex survey design.

### Findings

Twenty-two percent of our sample experienced racial discrimination in a healthcare setting in the past year. Discrimination was more common among those who had ever used MMT ($x^2$ = 10.00, p = 0.001) and racial/ethnic minorities ($x^2$ = 23.15, p<0.001). The interaction effect was much stronger than the main effects of race/ethnicity and MMT status. MMT status (versus no MMT) was positively associated with discrimination among Blacks (aOR = 3.93, 95% CI = 3.87–3.98, p<0.001), Whites (aOR = 2.25, 95% CI = 2.23–2.27, p<0.001),

**Data Availability Statement:** The National Epidemiologic Survey on Alcohol and Related Conditions-III (NESARC-III) data cannot be shared because its use is restricted by the National

Institute of Alcohol Abuse and Alcoholism (NIAAA). Access to the dataset is granted by NIAAA to researchers who successfully apply. Once obtained, NIAAA does not grant dataset users permission to share NESARC-III data. The authors of the current study did not have any special access privileges to the dataset that other applicants would not have. Interested researchers will be able to replicate the results of this study by following the protocol outlined in the current paper's Methods section. Researchers who are interested in obtaining the NESARC-III dataset through NIAAA can find more information here: https://www.niaaa.nih.gov/research/nesarc-iii/nesarc-iii-data-access.

**Funding:** The authors received no specific funding for this work.

**Competing interests:** The authors have declared that no competing interests exist.

and Latino/Latinas (aOR = 1.59, 95% CI = 1.55–1.62, p<0.001). Among American Indian/Alaska Natives (AI/AN), those who had used MMT had over thirty times the odds of racial discrimination, compared to their non-MMT counterparts (aOR = 32.78, 95% CI = 31.16–34.48, p<0.001).

## Conclusion

Race/ethnicity alone did not sufficiently account for racial discrimination in healthcare settings among those with a lifetime OUD. MMT status was strongly associated with racial discrimination among AI/AN. Our strong interaction effect is indicative of an additional barrier to health services utilization among AI/AN, which has important implications for OUD treatment outcomes and comorbidities. Health promotion programs aimed at increased adoption of MMT are promising, but should be considered in the context of racial/ethnic disparities, drug use and MMT stigma, and implicit biases in clinical settings.

## Background

In the U.S., the prevalence of heroin and opioid misuse has risen substantially over the past decade [1], disproportionately impacting Whites [2, 3], women [4], and less wealthy and underinsured populations [5]. Opioid misuse is commonly linked to community availability of opioids, perceptions of harms, and structural factors such as economic hardship [6, 7], but the causal linkages may be bidirectional for many of these factors.

Broadly, the racial/ethnic and gender trends in opioid misuse closely mirror those observed in the contemporaneous rise in heroin- and opioid-related overdose deaths [8, 9]. However, rates of overdose deaths are actually higher among non-Whites in some U.S. states. For example, the rates of overdose deaths among Blacks are more than ten points higher than Whites in Iowa (White, 7 per 100,000; Black, 21 per 100,000), Missouri (White, 16 per 100,000; Black, 32 per 100,000), and Washington D.C. (White, 9 per 100,000; Black, 60 per 100,000) [10]. Relatedly, the rate of change in overdose deaths also varies by state and by race/ethnicity. For example, in Minnesota, the rate of change has been the greatest among American Indian/Alaska Natives (AI/AN) (123% increase), followed by Blacks (105%) and Whites (72%) [11]. Opioid use is also positively associated with arrest and incarceration [12–14], resulting in the overrepresentation of opioid use disorders (OUDs) in incarcerated populations [15].

Methadone maintenance therapy (MMT) is one type of a broad range of medication-assisted therapies (MAT), and many research reports vary in terms of the specific type of medication assistance being studied. While the focus of the current study is MMT, a general understanding of other types of MAT is useful in developing a broad understanding of disparities in OUD treatment. MMT for the treatment of OUD is effective in reducing withdrawal symptoms and maintaining abstinence [16]. Medication-assisted therapies in general, and MMT specifically, are positively associated with better health and social outcomes across groups, including a decreased frequency of injecting opioids, less involvement in criminal activity, a decrease in infectious disease exposure, improved work productivity and employment, and other functional outcomes [17–19]. The use of MMT has increased steadily since the early 2000s [20], although reports of racial/ethnic disparities in MMT and MAT utilization are mixed. For example, Krawczyk and colleagues [21] found that White heroin users were the least likely to receive medication-assisted treatment (either methadone or buprenorphine),

even though Whites have demonstrated the highest prevalence of OUD in recent years. Conversely, other findings have illustrated a lower likelihood of both MMT-specific and MAT uptake among Blacks and Hispanics [22, 23].

In the U.S., experiences of racial discrimination in healthcare settings are more common among racial/ethnic minorities than among Whites. [24, 25]. MMT status—itself a source of stigma and bias—may exacerbate racial discrimination and impact treatment retention [26]. In a subsample of people who use illicit drugs, McKnight and colleagues [27] found that perceptions of institutional racial discrimination in healthcare settings negatively affected health services utilization, and that Latino/Latina drug users were the least likely to perceive institutional racism. However, the authors did not disaggregate their findings by opioid-specific substances or participation in MMT. In addition to racial/ethnic discrimination, stigmas toward drug use and opioid-based medication treatment are also common experiences among MMT participants [28]. Drug use stigma is a barrier to substance use treatment utilization among MMT participants, and the cumulative stigmas of age, race, and poverty compound this relationship [29]. Furthermore, healthcare professionals providing non-OUD services to MMT clients are largely untrained in addiction medicine, which plays a role in the perpetuation of stigma towards OUDs in healthcare and clinical settings [30].

Importantly, studies on racial/ethnic differences in the effects of MMT on discrimination are lacking in the public health literature. However, several studies in MMT settings have illustrated the importance of focusing on racial discrimination. For example, Black and Latino/Latina patients are more likely than Whites to receive methadone doses lower than the recommended amount [31, 32]. In considering differences in MMT-related factors between racial/ethnic groups, AI/AN have been largely excluded from MMT research. However, and of particular importance to this study, AI/AN MMT clients have cited institutional discrimination and experiences of prejudice in non-Indian Health Service settings as barriers to accessing and complying with MMT services [33].

Volkow and colleagues [34] recently and thoughtfully called for greater involvement of healthcare professionals in improving access to medication-assisted therapies and addressing comorbidities. However, this insight into the future of opioid treatment should also include consideration of possible negative MMT outcomes. Racial discrimination is common in general healthcare settings, but there may be underlying mechanisms driving discriminatory behavior in addition to race/ethnicity. MMT status may compound healthcare-based racial discrimination. In the current study, we sought to identify group differences in the association between MMT status and experiences of racial discrimination in healthcare settings between AI/ANs, Blacks, Latino/Latinas, and Whites with lifetime heroin or opioid use disorders. We hypothesized that 1) MMT status would be positively associated with racial discrimination, 2) racial/ethnic minority statuses would each be positively associated with racial discrimination and group odds ratio estimates would vary in magnitude, compared to Whites, and 3) the interaction effects would be stronger than the main effects alone.

## Methods

### Data source

We used the National Survey on Alcohol and Related Conditions-III (NESARC-III) (2012–2013). A general description of NESARC-III and data collection methods have been described elsewhere [35]. In short, NESARC-III provides a nationally representative sample with variables describing substance use disorders and treatment services utilization, as well as many social and environmental indicators related to public health. The dataset includes estimates for

population weights and strata, reflective of the parent study's complex sampling methods. The household survey response rate was 72%. Analyses of complex survey data requires the use of weights to compensate for variable probabilities of selection and differential nonresponses rates by group. NESARC-III provides pre-calculated sampling weights designed to calibrate the sampling distribution for major subgroups defined by region, sex, age, and race/ethnicity [35]. In the full, unrestricted dataset (survey n = 36,309; weighted population n = 235,411,957), less than 0.1% of those without an OUD had a record of MMT participation. Thus, in order to focus on MMT participants as a subset of those with OUD, we restricted the dataset to include only individuals with records indicating a lifetime OUD (inclusive of both heroin use disorder and opioid use disorder) and complete data for all study variables (survey n = 766; weighted population n = 5,276,507).

## Variables

Our outcome of interest was a binary indicator of whether an individual experienced racial discrimination in a healthcare setting in the past year (yes or no [referent]). Response options to discrimination survey questions included very often, fairly often, sometimes, almost never, and never. Our binary variable included those who endorsed no experience (never) or any experience (very often, fairly often, sometimes, or almost never) with racial discrimination in a healthcare setting. Survey respondents who reported any experience with either of the following two questions were coded as having ever experienced racial discrimination in a healthcare setting: 1) During the last 12 months, about how often did you experience discrimination in your ability to obtain healthcare or health insurance coverage because of your race or ethnicity? 2) During the last 12 months, about how often did you experience discrimination in how you were treated when you got care because of your race or ethnicity? The type of healthcare setting was not specified in these two NESARC-III questions, leaving the interpretation of what constitutes a healthcare setting up to the survey respondent.

The main effects of interest were MMT treatment utilization and race/ethnicity. MMT status included whether an individual ever attended MMT treatment services (ever or never [referent]). We did not consider past-year MMT utilization alone due to small sample size and subsequent regression parameter estimation problems caused by small covariate cell sizes. Also, NESARC-III does not capture utilization of other agonist therapies such as buprenorphine. Our racial/ethnic groups included White (referent), Black, Latino/Latina, and American Indian/Alaska Native. We did not include the Asian/Native Hawaiian/Other Pacific Islander group in our analyses because of its small sample size, and for whom several cross-tabulation cells were equal to zero.

We also considered several covariates in our study, based on *a priori* knowledge of factors that likely confound the relationships between MMT, race/ethnicity, and discrimination in healthcare settings. We included sex (male [referent] or female), age group (18–29 [referent], 30–39, 40–49, 50+ years old), geography (urban [referent] or rural), sexual orientation minority status (heterosexual/straight [referent], or gay/lesbian, bisexual, or not sure), opioid use disorder type (heroin use disorder only [referent], illicit/prescription opioid use disorder only, or both), past-year dual mental health disorder (none [referent] or any), ever homeless in the past year (never [referent] or ever), enrollment in Medicare or Medicaid (yes or no [referent]), and any history of incarceration (yes or no [referent]). We also considered past-year non-opioid substance use disorder diagnoses (none [referent], alcohol use disorder only, drug use disorder only, or both alcohol and drug use disorders), using pre-defined diagnostic variables provided by NESARC which are based on criteria for substance use disorders outlined by the Diagnostic and Statistical Manual of Mental Disorders, Fifth Edition.

## Analysis

We used SAS software (v9.4) [36] and applied survey weights to all analyses. We used multi-variable logistic regression to model racial discrimination in healthcare settings. Specifically, we used an interaction term between MMT and race/ethnicity to calculate the associations between MMT and racial discrimination within racial/ethnic groups. Within-group adjusted odds ratios, 95% confidence intervals, and p-values were generated using LSMEANS and SLICEDIFF options using PROC GLIMMIX. For our fitted model, following a strategy of purposeful selection of covariates for multivariable regression outlined by Hosmer and Lemeshow [37], we carried forward variables with chi-square p-values less than 0.20 from bivariate tests.

## Results

The majority of individuals with a lifetime OUD were White (79%), younger (31% 18–29 years, 22% 30–39 years), resided in rural areas (75%), and were not covered by Medicare or Medicaid (66%) (Table 1). A clear minority ever utilized MMT (9%), and nearly half had a co-occurring substance use disorder (44%) and had ever been incarcerated (46%).

Twenty-two percent of those with a lifetime OUD experienced racial discrimination in a healthcare setting in the past year. Experiences of racial discrimination were more likely among those who had ever used MMT (p = 0.002), racial/ethnic minority groups (p<0.001), men (p = 0.023), those with a past-year dual disorder diagnosis (p<0.001), those who were homeless in the past year (0.063), those enrolled in Medicare or Medicaid (p = 0.019), and those with a history of incarceration (p = 0.006).

Table 2 shows results from our fitted model. Only sex, past-year dual disorder diagnosis, past-year homeless status, enrollment in Medicare/Medicaid, and incarceration history were carried forward from bivariate analyses. Individuals with a lifetime OUD who had ever utilized MMT had more than four times higher odds of experiencing racial discrimination in healthcare settings, compared to their non-MMT counterparts (adjusted odds ratio [aOR] = 4.28, 95% confidence interval [95% CI] = 4.21–4.35, p<0.001). All racial/ethnic minority groups had higher odds of discrimination than Whites, and the association was particularly pronounced among AI/AN (aOR = 4.99, 95% CI = 4.86–5.12, p<0.001) and Blacks (aOR = 3.93, 95% CI = 3.87–3.98, p<0.001).

The magnitude of associations between MMT and racial discrimination in healthcare settings varied widely between racial/ethnic groups. Among Blacks, Whites, and Latino/Latinas, those who ever utilized MMT (versus never) had higher odds of experiencing racial discrimination (Black—aOR = 2.87, 95% CI = 2.79–2.94, p<0.001), (White—aOR = 2.25, 95% CI = 2.23–2.27, p<0.001), (Latino/Latina—aOR = 1.59, 95% CI = 1.55–1.62, p<0.001). The odds of racial discrimination were especially pronounced among AI/AN, such that those who had ever used MMT demonstrated more than 30 times higher odds of discrimination, compared to their non-MMT counterparts (aOR = 32.78, 95% CI = 31.16–34.48, p<0.001).

With respect to other covariates of interest, women demonstrated lower odds of experiencing racial discrimination in healthcare settings than men (aOR = 0.51, 95% CI = 0.50–0.52, p<0.001). Those with past-year dual disorder diagnoses (aOR = 3.38, 95% CI = 3.36–3.40, p<0.001), had been homeless in the past year (aOR = 1.21, 95% CI = 1.20–1.22, p<0.001), were enrolled in Medicare/Medicaid (versus not enrolled) (aOR = 1.35, 95% CI = 1.34–1.36, p<0.001) and had a history of incarceration (versus no history) (aOR = 1.29, 95% CI = 1.28–1.30, p<0.001) had higher odds of experiencing racial discrimination in a healthcare setting.

**Table 1. Weighted descriptive characteristics of individuals with lifetime opioid use disorders (NESARC-III, 2012–2013) (survey n = 766; weighted population n = 5,276,507).**

| Variables | Total | Past-year experience with racial discrimination in healthcare settings | | $x^2$ | p |
| --- | --- | --- | --- | --- | --- |
| | | Ever experienced 22.48% | Never experienced 77.52% | | |
| | % | col % | col % | | |
| **Methadone treatment status** | | | | 10.00 | 0.002 |
| Never utilized | 90.93 | 83.86 | 92.95 | | |
| Ever utilized | 9.07 | 16.13 | 7.04 | | |
| **Race/ethnicity** | | | | 23.15 | <0.001 |
| White | 78.86 | 64.45 | 83.05 | | |
| Black | 9.21 | 17.01 | 6.94 | | |
| Latina/Latino | 9.29 | 14.27 | 7.84 | | |
| American Indian/Alaska Native | 2.64 | 4.27 | 2.17 | | |
| **Sex** | | | | 5.04 | 0.023 |
| Male | 54.05 | 62.91 | 51.48 | | |
| Female | 45.94 | 37.09 | 48.51 | | |
| **Age group** | | | | 2.46 | 0.482 |
| 18–29 | 30.97 | 25.52 | 32.55 | | |
| 30–39 | 21.64 | 23.01 | 21.25 | | |
| 40–49 | 19.05 | 19.62 | 18.89 | | |
| 50+ | 28.33 | 31.85 | 27.30 | | |
| **Urbanicity** | | | | 0.43 | 0.510 |
| Urban | 22.57 | 24.91 | 21.89 | | |
| Rural | 77.42 | 75.09 | 78.10 | | |
| **Sexual orientation minority** | | | | 0.00 | 0.983 |
| No | 91.72 | 91.67 | 91.73 | | |
| Yes | 8.28 | 8.32 | 8.27 | | |
| **Opioid use disorder type** | | | | 0.26 | 0.880 |
| Opioid use disorder only | 79.39 | 80.98 | 78.93 | | |
| Heroin use disorder only | 9.81 | 8.84 | 10.10 | | |
| Both opioid and heroin use disorders | 10.78 | 10.17 | 10.96 | | |
| **Past-year non-opioid substance use disorder diagnoses** | | | | 2.45 | 0.458 |
| None | 55.64 | 51.87 | 56.74 | | |
| Alcohol use disorder only | 23.28 | 26.40 | 22.37 | | |
| Drug use disorder only | 8.14 | 10.72 | 7.39 | | |
| Both AUD and DUD | 12.94 | 11.00 | 13.50 | | |
| **Past-year dual disorder diagnosis** | | | | 23.34 | <0.001 |
| None | 40.45 | 22.76 | 45.59 | | |
| Any | 59.54 | 77.24 | 54.41 | | |
| **Ever homeless in the past year** | | | | 3.46 | 0.063 |
| No | 90.07 | 85.22 | 91.49 | | |
| Yes | 9.92 | 14.78 | 8.51 | | |
| **Medicare or Medicaid coverage in the past year** | | | | 5.54 | 0.019 |
| No | 65.54 | 56.45 | 68.18 | | |
| Yes | 34.45 | 43.55 | 31.82 | | |
| **Ever been incarcerated** | | | | 7.64 | 0.006 |
| No | 53.71 | 42.52 | 56.97 | | |
| Yes | 46.29 | 57.48 | 43.03 | | |

Notes: Percentages are weighted, column sums may not equal 100.00; Rao-Scott $x^2$

**Table 2. Multivariable logistic regression modeling past-year experience with racial discrimination in healthcare settings among individuals with lifetime opioid use disorders (NESARC-III, 2012–2013) (survey n = 766; weighted population n = 5,276,507).**

| Variables | aOR | 95% CI | p |
|---|---|---|---|
| *Main effects* | | | |
| **MMT status** | | | |
| Never utilized MMT | 1.00 | | |
| Ever utilized MMT | 4.28 | (4.21, 4.35) | <0.001 |
| **Race/ethnicity** | | | |
| White | 1.00 | | |
| Black | 3.93 | (3.87, 3.98) | <0.001 |
| Latino/Latina | 2.16 | (2.13, 2.19) | <0.001 |
| American Indian/Alaska Native | 4.99 | (4.86, 5.12) | <0.001 |
| *Interaction effects* | | | |
| **MMT status * Race/ethnicity** *Within-group comparisons of MMT status (Ever vs. Never [ref])* | | | |
| White | 2.25 | (2.23, 2.27) | <0.001 |
| Black | 2.87 | (2.79, 2.94) | <0.001 |
| Latino/Latina | 1.59 | (1.55, 1.62) | <0.001 |
| American Indian/Alaska Native | 32.78 | (31.16, 34.48) | <0.001 |
| *Covariates* | | | |
| **Sex** | | | |
| Male | 1.00 | | |
| Female | 0.51 | (0.50, 0.52) | <0.001 |
| **Past-year dual disorder diagnosis** | | | |
| None | 1.00 | | |
| Any | 3.38 | (3.36, 3.40) | <0.001 |
| **Ever homeless in the past year** | | | |
| No | 1.00 | | |
| Yes | 1.21 | (1.20, 1.22) | <0.001 |
| **Medicare or Medicaid coverage in the past year** | | | |
| No | 1.00 | | |
| Yes | 1.35 | (1.34, 1.36) | <0.001 |
| **Ever been incarcerated** | | | |
| No | 1.00 | | |
| Yes | 1.29 | (1.28, 1.30) | <0.001 |

## Discussion

We found evidence of a strong interaction effect of race/ethnicity on the association between MMT and racial discrimination in healthcare settings. Previous research has demonstrated that MMT is effective in treating OUDs, with decades of research documenting positive clinical and social outcomes among diverse patient populations. However, MMT status may also have negative consequences. We sought to better understand racial/ethnic differences in the association between MMT and experiences of racial discrimination in general healthcare settings. Specifically, we found that MMT status was independently and positively associated with experiences of racial discrimination in our adjusted model. Among AI/AN, the association between MMT status and experiences of racial discrimination was positive and very strong.

Reports on disparities in MMT outcomes are rare. Rather, disparities research tends to focus on antecedents to MMT utilization, including barriers to treatment initiation and

implicit bias in treatment and healthcare settings. Our findings contribute to the MMT literature by framing discrimination as an unintended consequence associated with MMT and by focusing on differential racial/ethnic effects of MMT-related discrimination. Our study complements a recent meta-analysis by Fitzgerald and Hurst [38], who demonstrated occasions of implicit bias among healthcare providers, ranging from predominately race-based biases to biases against intravenous drug users, patients with mental health problems, and people with multiple marginalizing characteristics. Our study also aligns with work by Hansen and Roberts [39], who identified that MMT is often portrayed using politicized imagery of poor, Black, and Hispanic treatment-seekers. In stark contrast to the stigma surrounding MMT, Lagisetty and colleagues [40] recently demonstrated that buprenorphine is associated with White treatment clients who pay with private insurance or self-pay for services. We expand on the current literature by measuring the compounding effects of MMT and race/ethnicity on racial discrimination. In this sense, race/ethnicity alone does not sufficiently explain race-based biases in healthcare settings.

We found that women were less likely than men to perceive racial discrimination in healthcare settings. Rowan-Szal and colleagues [41] found that women in MMT were more likely than men to build positive rapport with clinic staff, and women generally demonstrated higher levels of motivation to successfully complete treatment. Relatedly, women are more likely than men to remain in MMT services for up to 90 days [42], while several other studies have demonstrated better MMT outcomes among women [43, 44]. Despite the broad gender differences in MMT outcomes, more research is needed to identify gender differences within racial/ethnic groups. In addition, experiences of racial discrimination may also vary by gender when stratified by racial/ethnic group. Future research is needed to identify disparate trends in discrimination and MMT by both race/ethnic and gender groups.

MMT clients have complex healthcare needs and typically present with multiple comorbidities [45, 46]. For example, results from our descriptive analyses indicated that nearly 60% of OUD treatment participants also had a dual mental health diagnosis in the past year. This finding was expected and aligns with previous reports that dual mental health and psychiatric disorders are exceptionally high among individuals with OUD [47, 48]. Importantly, additional research has pointed to the intersectional relationships between mental health problems, MMT utilization, and the experience of social inequities, in particular among MMT patients who also struggle with mental health diagnoses [49]. Future research may begin to unpack the complex relationships between race/ethnicity, mental health, stigma, discrimination, and MMT services utilization.

The interaction between MMT and race/ethnicity highlights an additional barrier to receiving general health services. Experiences of discrimination negatively affect compliance and further utilization of general healthcare [24, 50], which may worsen MMT comorbidities by delaying or foregoing participation in other specialty services. Specifically, AI/AN have demonstrated poorer health on multiple metrics relative to their non-AI/AN counterparts [51], which may be partially due to the convergence in AI/AN communities of the opioid epidemic, multiple barriers to behavioral and general health services, and the under-treatment of comorbid conditions. The treatment and resolution of comorbidities is effective when addressed in parallel to the treatment of OUD or in integrated behavioral health clinical settings [52, 53] but racial discrimination by health providers is associated with low health services utilization [24]. Thus, it is reasonable to infer that resolving co-occurring health problems in integrated healthcare settings is a critical yet particularly challenging obstacle to OUD treatment success in general, and success among AI/AN in particular.

Our findings also align with recent calls to incorporate AI/AN subsamples into research addressing opioid- and treatment-related outcomes. While the Indian Health Service (IHS)

has made recent and notable strides in improving OUD treatment outcomes [54, 55], research into determinants of treatment outcomes among AI/AN has not kept up. While the implementers of the NESARC-III survey worked closely with tribal leaders to reach a nationally representative sample of AI/AN participants [35], it was not possible to discern whether MMT was prescribed through the IHS in this dataset. However, given the IHS' emphasis on culturally-centered health services [56], as well as AI/ANs' preferences for IHS-specific health services and the incorporation of traditional indigenous elements into health services delivery [57, 58], it is reasonable to suggest that our strong findings of racial/discrimination in healthcare settings among AI/AN may be driven, in part, by AI/AN biases inherent in non-IHS services [51, 59]. Importantly, the majority of AI/AN do not receive health services through the IHS [56], and only 1% of substance abuse treatment providers identify as AI/AN [60]. Given this relatively low coverage of IHS services, the risk for experiences of discriminatory behavior may be higher among AI/AN than other groups. Broadly, the findings from this study may inform future research into treatment services within AI/AN communities. For example, scientific investigations may be warranted that address components of the ongoing National Institutes of Health initiative entitled Interventions for Health Promotion and Disease Prevention for Native American Populations. These programs are specifically aimed at increasing health services utilization and improving health through 1) community-driven intervention and research processes, 2) multi-sectoral research and practice partnerships, and 3) support for emerging tribal research infrastructures [61].

## Limitations

NESARC-III only includes information on MMT—there are no indicators for other medication-assisted therapies (buprenorphine, naltrexone) that are also of interest in opioid treatment outcomes research. The association between MMT and racial discrimination likely differs depending on the specific type of medication assistance. In addition to methadone, future research may compare differential effects of buprenorphine and naltrexone on OUD treatment outcomes.

NESARC-III data is cross-sectional in nature, limiting interpretations of study results to associations. We cannot infer causal relationships with this data. Also, the NESARC-III sample includes only survey participants based in the US. While OUD and MMT treatment disparities may be widespread globally, we cannot generalize our findings to other global settings. Finally, NESARC-III data ranges from 2012–2013. There have been multiple public health and policy initiatives aimed at preventing OUD and reducing the harm associated with it since the collection of NESARC-III data. Thus, it is possible that a replicated study using more current data may show differences in the size and strength of associations.

## Conclusion

MMT is an effective OUD treatment strategy, but there remains a need to investigate negative MMT outcomes. We found that MMT status is associated with experiences of racial discrimination in general healthcare settings. Race/ethnicity moderated this relationship, such that AI/AN who had ever used MMT had drastically higher odds of past-year discrimination, compared to the effect of MMT within all other racial/ethnic groups. Our findings highlight racial discrimination as a possible barrier to healthcare services utilization among groups with multiple marginalized statuses—namely AI/AN with a history of OUD and MMT use. Furthermore, it is important to note the challenges of interpreting one expression of bias (racial discrimination) in the midst of other latent factors that may also affect how racial discrimination is both expressed and perceived in healthcare settings.

Promoting culturally tailored healthcare may reduce implicit biases towards underrepresented patient populations. Integrating mental and behavioral health services into general medical practices may strengthen providers' understanding of addiction and treatment, as well as their ability to treat OUD-related comorbidities. Health promotion interventions should continue the focus on increasing MMT uptake, while also working to dissolve many of the underlying systemic and discriminatory barriers to treatment and other health services utilization.

## Author Contributions

**Conceptualization:** George Pro, Nick Zaller.

**Data curation:** George Pro.

**Formal analysis:** George Pro.

**Investigation:** George Pro, Nick Zaller.

**Methodology:** George Pro, Nick Zaller.

**Resources:** George Pro.

**Software:** George Pro.

**Supervision:** George Pro.

**Validation:** Nick Zaller.

**Writing – original draft:** George Pro.

**Writing – review & editing:** Nick Zaller.

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
