## [Decision Letter · Decision Letter 0]

25 Jul 2019

PONE-D-19-17408

Interaction effects in the association between methadone maintenance therapy and experiences of racial discrimination in healthcare settings

PLOS ONE

Dear Dr. Pro,

Thank you for submitting your manuscript to PLOS ONE. After careful consideration, we feel that it has merit but does not fully meet PLOS ONE’s publication criteria as it currently stands. Therefore, we invite you to submit a revised version of the manuscript that addresses the points raised during the review process.

The manuscript which you submitted to PLOS ONE, has been reviewed. The reviewers comments are included at the bottom of this letter. The reviewers would like to see major revisions made to your manuscript before the final decision. Therefore, I invite you to respond to the reviewers comments as soon as posible.

We would appreciate receiving your revised manuscript by Sep 08 2019 11:59PM. To enhance the reproducibility of your results, we recommend that if applicable you deposit your laboratory protocols in protocols.io, where a protocol can be assigned its own identifier (DOI) such that it can be cited independently in the future. For instructions see: http://journals.plos.org/plosone/s/submission-guidelines#loc-laboratory-protocols

We look forward to receiving your revised manuscript.

Kind regards,

**Rodrigo Marín-Navarrete, Ph.D.**

Academic Editor

PLOS ONE

**Journal Requirements**

2. We noticed that you have chosen the subsection category “[FOR JOURNAL STAFF USE ONLY]” for your manuscript. Unfortunately, this is not a valid category. At this time, please choose one or more subsections that best represent the topic(s) of your study.

**Comments to the Author**

1. Is the manuscript technically sound, and do the data support the conclusions?

Reviewer #1: Partly

Reviewer #2: Yes

Reviewer #3: Yes

2. Has the statistical analysis been performed appropriately and rigorously? 

Reviewer #1: Yes

Reviewer #2: Yes

Reviewer #3: Yes

3. Have the authors made all data underlying the findings in their manuscript fully available?

Reviewer #1: No

Reviewer #2: Yes

Reviewer #3: Yes

4. Is the manuscript presented in an intelligible fashion and written in standard English?

Reviewer #1: Yes

Reviewer #2: Yes

Reviewer #3: Yes

5. Review Comments to the Author

Reviewer #1: Thank you for the invitation to review this interesting manuscript.

The issue of discrimination, including racial discrimination, and its relationship with access to quality health care is an important one, and the NESARC dataset is a comprehensive and well-respected source. I offer the following comments in the hope of assisting the authors in fine tuning the manuscript.

The major source of non-clarity in the manuscript is the assumption that participation in MMT is the source of the discrimination being reported by people. In this paper, lifetime MMT participation is being effectively used as a proxy for having, or having had, an opioid use disorder (OUD). The discussion of drug-based stigma in the manuscript does not explain this and so is somewhat convoluted with discussion moving back and forth between the related but not equivalent issues of drug use, MMT and OUD. There is a literature that discusses the stigma attached to substance use disorders, and the idea that intra-group discrimination exists such that people who inject drugs (PWID) are discriminated against by other users of illicit drugs. Those who have ever participated in MMT are likely to be a subset of PWID and/or a subset of people with OUD.

*In order to differentiate between MMT participation per se and OUD, the author’s model would need to have adjusted for OUD, rather than excluding it from the covariates chosen.* I believe this is a significant issue with the model presented.

The links being sought between a lifetime experience of MMT and racial discrimination make less sense than potential links between identification of a person as someone with an OUD, or a PWID, and racial-based discrimination. It is also possible that the discrimination perceived as racial may in fact be an expression of substance-based discrimination – this concept is also relevant but not explored. There is also a literature that discusses the bi-directional nature of the relationship between substance use and discrimination. This is particularly important for this manuscript, as the relationships between discrimination and MMT participation/uptake are, rightly, of concern.

1. Background section

The literature discussed is entirely North American. This should be acknowledged in the text; differences in patterns of use and misuse vary world-wide.

Language used should be precise to avoid confusion, particularly in discussions of complex, inter-related topics. A number of examples are given below.

The term ‘drug abuse’ is not generally accepted in other regions. ‘Misuse’ is considered more acceptable.

The word ‘prevalence’ should be used in place of ‘incidence’ for heroin and opioid misuse, as the paper referenced does not measure incidence.

Similarly, poly substance use is a risk for overdose, rather than opioid misuse; the factors listed are linked to prevalence of misuse, rather than attributed to those factors, s the causal linkages may be bidirectional for many of them.

Paragraph 2 starts talking about rates of OD deaths, but gives an example of the rate of change in the number of OD deaths. These concepts are related but not interchangeable.

Similarly in paragraph 3, the authors switch between discussions of MMT (specifically methadone) and other MATs (medication assisted treatments, which include other OSTs).

In paragraph 4, “experiences of racial discrimination ….more common… racial/ethnic minorities”… but a comparison group is not cited.

McNight’s paper apparently found that discrimination differed between groups, but which groups?

In paragraph 5 (line 87) is it possible the text should read “differences in the effects on MMT of discrimination? The following sentence/s refers to the link between race and receipt of appropriate MMT.

In the final paragraph of the Background, it is somewhat tautological that racial/ethnic minority status would be associated with racial discrimination.

Methods:

Variables in NESARC describe substance use disorders etc, rather than ‘addressing’ them. Stratum is a singular noun – perhaps should be strata.

In line 136, perhaps use gender rather than ‘sex’, to avoid confusion with ‘sexual minority status’, which is presumed to refer to sexual orientation.

In the Analysis section, why was a cut-off of 0.2 used for the p-values for inclusion in the model, rather than the usual 0.05?

Results:

Does Table 2 represent a single multivariable model, adjusted for all listed variables and covariates? If so, why does line 200 in the Discussion refer to ‘models’?

Discussion:

In general, a note that the relationships explored in this analysis are associations, and causality cannot be assumed. Therefor (line 201) one variable should not be discussed as having an effect on another.

Line 208: the Fitzgerald article describes occasions of bias, rather than trends (changes).

Line 210: …”people with marginalized characteristics” should read "marginalizing characteristics"

Paragraph 3 of the Discussion draws some long bows without references, e.g. the experience of discrimination negatively affecting compliance (there are references for this) and the “worsening of MMT comorbidities” (unspecified) which may refer to comorbid conditions going untreated if people avoid healthcare in general after experiencing discrimination. Line 223 may read more clearly as “..the convergence in AI/AN communities of the opioid epidemic, multiple barriers… “ etc as the ‘opioid epidemic’ is not specific to those communities.

The final statements in this paragraph seem to suggest that treatment of conditions comorbid with OUDs must precede treatment for the OUDs: this is an outdated concept. There is considerable literature suggesting that it is more effective to address multiple co-occurring conditions in parallel or in integrated fashion, rather than sequentially.

The final comments relate to challenges faced in service delivery for AI/AN communities. There is something of a leap in stating that the findings from this study justify initiatives targeting Native American populations, however essential those initiatives are.

The Limitations section should include a statement noting that relationships cannot be construed as causal, and that data are from the USA only and may not generalise to other settings.

The Conclusion refers to issues that are worthy of attention but extrapolates somewhat from the findings, as it appears to infer that it is racial discrimination that is the major barrier to health service uptake, particularly OST, rather than this as perhaps an expression of the multiple other stigma associated with illicit drug use.

Finally - some attention should be paid to accuracy and consistency of formatting of the references.

Reviewer #2: This is article represents an important contribution to the literature on discrimination of patients with OUD. This paper demonstrates independent effect of Methadone Maintenance Therapy on racial / discrimination in general health care settings. Authors do a great job in explaining the objectives and the methods to support their hypotheses.

Some minor comments:

Background:

The authors could include some of the following references as part of the citations of the effects on the positive effects of medication-assisted treatment.

Ma J, Bao YP, Wang RJ, et al. Effects of medication-assisted treatment on mortality among opioids users: a systematic review and meta-analysis. Molecular psychiatry. 2018.

Maglione MA, Raaen L, Chen C, et al. Effects of medication assisted treatment (MAT) for opioid use disorder on functional outcomes: A systematic review. Journal of substance abuse treatment. 2018;89:28-51.

Methods:

It would be important to explain / operationalize racial discrimination in healthcare settings- do these include discrimination in all types of health care settings including substance abuse specialty treatment centers, that is centers where methadone might be dispense or does it exclude it? A clear explanation of what healthcare settings includes can help in interpreting the findings and understanding their full implication.

Discussion:

A stronger justification towards the findings in AI/AN is warranted.The statement “our strong findings of racial/discrimination in healthcare settings among AI/AN may be driven, in part, by AI/AN biases inherent in non-IHS services” requires a citation and a broader explanation to contextualize the findings.

Limitations:

Authors should acknowledge that the dataset ( NESARC- III) is from 2012-2013, and while racial discrimination in healthcare settings by individuals on MMT might be pervasive and not easy to change, mulitple national initiatives towards the opioid epidemic and education towards evidence based treatments within mainstream of care might shift the results observed, should we look at current data.

Reviewer #3: The paper could be accepted. However, a major review must be done and after that should be reviewed again.

This paper provides some evidence on the role of race and ethnicity on discrimination among opiate-dependent patients in MMT. Twenty-two percent of the sample experienced racial discrimination in a healthcare setting. This is a relevant an a not well-studied topic and the findings of the paper should be taking account for the future. However, there is a lack of some relevant factors that must be detail and explain.

General comments

There is some typos/ deleted references f.e MMT vs. no MMT [ref[), see all the mns.

The expression “mental health disorder diagnosis” should be change by Dual Disorders, in the text, tables, etc for a review of the use of this term see Szerman et al. Rethinking Dual disorders/pathology. Addictive Disorders & Their Treatment. 2013; 12 (1): 1–10 doi: 10.1097/ADT.0b013e31826e7b6a

It is necessary to use the term Latino/Latina? Or Latino includes all?

Abstract

It is clear according the result and discussion, but should be modified if there is new comments or results.

Introduction

Read: McKnight et al. Perceived discrimination among racial and ethnic minority drug users and the association with health care utilization. J Ethn Subst Abuse. 2017;16(4):404-419. doi: 10.1080/15332640.2017.1292418.

Methods & Results:

Page 6 “Our outcome of interest was a binary indicator of whether an individual experienced 120 racial discrimination in a healthcare setting in the past year (yes or no). Response options to 121 discrimination survey questions included very often, fairly often, sometimes, almost never, and 122 never. Our binary variable included those who endorsed no experience (never) or any experience…”

I suppose that that no racial discriminations means that the answer is “never” and yes means “very often, fairly often, sometimes, almost never”, in any of the questions, please clarify this sentence.

Could you show if there are differences between patients on MMT coming from heroin-dependence or opiate pain-killer dependence?

Discussion

I think that a short sentence summarizing all your relevant results it will do the reading more clear. After that you can discuss your results.

The presence of Dual disorders is high as expected is close to 60% last year, according other research among opiate-dependent patients see Roncero et al. Psychiatric comorbidities in opioid-dependent patients undergoing a replacement therapy programme in Spain: The PROTEUS study. Psychiatry Res. 2016 Sep 30;243:174-81. doi:10.1016/j.psychres.2016.06.024. and Martínez-Luna et al. Harm reduction program use, psychopathology and medical severity in patients with methadone maintenance treatment. Adicciones. 2018 Jan 15;30(3):197-207. doi: 10.20882/adicciones.897.

Your results “Experiences of racial discrimination were more likely among …those with a past-year mental health) diagnosis (p<0.0001)”. I believe that is an important factor an any comments should be added on this topic.

Do you any date or any comments if there is an association between dual disorders / raze-ethnicity and discrimination??

Could you speculate if experiences of racial discrimination were more likely among any raze of dual disorders opiate-dependent? Should be done an special intervention or screening on discrimination in this patients?

It seems that there is no discrimination according sex, it is important to point out this result.

Could you provide any idea for the clinicians on the fact of how avoid or deal with this discrimination

Bibliography

I suggest read and consider to included in the mns this papers, that I have mentioned before.

6. PLOS authors have the option to publish the peer review history of their article (what does this mean?). If published, this will include your full peer review and any attached files.

Reviewer #1: No

Reviewer #2: No

Reviewer #3: No

---

## [Author Response · Author response to Decision Letter 0]

15 Aug 2019

Please also see our 'Response to Reviewers' document, uploaded as part of this resubmission.

Rodrigo Marín-Navarrete, PhD, Academic Editor

PLOS ONE

National Institute of Psychiatry

Mexico City, Mexico

Dear Dr. Marín-Navarrete

Thank you for providing us with an opportunity to revise our manuscript, “Interaction effects in the association between methadone maintenance therapy and experiences of racial discrimination in healthcare settings”. The feedback provided by our three reviewers was encouraging and insightful.

Broadly, our major revisions had to do with clarifying how our hypotheses informed our model construction and covariate selection, more clearly narrating the implications of our results in the context of our strong odds ratio estimates among American Indians/Alaska Natives in the discussion section, and summarizing suggestions for public health practice without extrapolating beyond our findings in the conclusion. 

Below we summarize our responses to the critiques. We have tracked changes in the revised manuscript and also provided a clean, untracked version (line numbers given throughout our response are in reference to the tracked changes version). Again, we thank you and the reviewers for improving the clarity and precision of our research. 

Sincerely,

George Pro, PhD (Corresponding Author) 

Postdoctoral Fellow 

Center for Health Equity Research 

george.pro@nau.edu
http://nau.edu/CHER

(928).523.4267

 

General feedback

Reviewer 1

The major source of non-clarity in the manuscript is the assumption that participation in MMT is the source of the discrimination being reported by people. In this paper, lifetime MMT participation is being effectively used as a proxy for having, or having had, an opioid use disorder (OUD). The discussion of drug-based stigma in the manuscript does not explain this and so is somewhat convoluted with discussion moving back and forth between the related but not equivalent issues of drug use, MMT and OUD. 

*In order to differentiate between MMT participation per se and OUD, the author’s model would need to have adjusted for OUD, rather than excluding it from the covariates chosen.* I believe this is a significant issue with the model presented.

Thank you for your thoughtful and insightful review. As our paper deals with multiple related factors around OUD, MMT, and racial discrimination, we want to be careful in articulating how our hypotheses informed our model creation and selection of covariates. 

Using NESARC-III data, we restricted the dataset to include only individuals who had a lifetime OUD (heroin use disorder or other opioid use disorder, n=766). Among our sample of individuals with any OUD, MMT participation is not necessarily a proxy indicator of OUD status, but rather a subset of those with an OUD. In this sense, OUD could not be excluded as a covariate, but instead defined our analytic sample. Our rationale for restricting our sample to those with OUD was that MMT participation would be unlikely among those without an OUD and somewhat of an anomaly in the dataset. Indeed, in a post-hoc analysis of the full, unrestricted dataset conducted for this review, less than 0.1% of those without an OUD received MMT. Thus, we restricted our dataset to include only those with an OUD, rather than adjusting for OUD using the full NESARC-III dataset. We have added a description of this rationale and the process we underwent to arrive at our final analytic sample in the methods section. (Lines 128-131) 

The links being sought between a lifetime experience of MMT and racial discrimination make less sense than potential links between identification of a person as someone with an OUD, or a PWID, and racial-based discrimination. It is also possible that the discrimination perceived as racial may in fact be an expression of substance-based discrimination – this concept is also relevant but not explored. 

We agree that intravenous opioid use is an important factor in the development of stigma and bias in healthcare settings and we appreciate the reviewer’s comment. We are discussing a future manuscript that addresses both racial and gender disparities in treatment outcomes among those who inject any illicit drug. 

While we believe that MMT can certainly be considered a proxy for intravenous opioid use or OUD more generally, we note that there is substantial stigma associated with MMT itself (i.e., it is often viewed as substituting one drug for another, and is often associated with a poorer class of drug users). Therefore, we believe that MMT is itself an important factor contributing to racial bias and thus warrants investigation separate from intravenous drug use and/or OUD. 

There is also a literature that discusses the bi-directional nature of the relationship between substance use and discrimination. This is particularly important for this manuscript, as the relationships between discrimination and MMT participation/uptake are, rightly, of concern.

We appreciate the feedback about the bidirectional nature of substance misuse and multiple factors related to discrimination and treatment services uptake. We have addressed this concern in our revisions, specifically in response to Comment #4. 

As suggested by a previous comment, a future study addressing the relationship between discrimination and MMT may reverse the independent and dependent variables used in the current study. For example, in such a follow-up study, we will specifically examine the interaction between discrimination (our current outcome) and race/ethnicity and its association with MMT (our current predictor) as an outcome. However, while important, the strategy of looking at multiple combinations of independent and dependent variables as suggested by this review was beyond the scope of the current study design. In our conclusion, we address many of the complexities inherent in interpreting our findings, particularly the challenge of interpreting one expression of bias (racial discrimination) in the midst of other latent factors that may also affect how racial discrimination is both expressed and perceived in healthcare settings.

Reviewer 2

This is article represents an important contribution to the literature on discrimination of patients with OUD. This paper demonstrates independent effect of Methadone Maintenance Therapy on racial / discrimination in general health care settings. Authors do a great job in explaining the objectives and the methods to support their hypotheses.

Thank you for this acknowledgement. 

Reviewer 3

The paper could be accepted. However, a major review must be done and after that should be reviewed again.

This paper provides some evidence on the role of race and ethnicity on discrimination among opiate-dependent patients in MMT. Twenty-two percent of the sample experienced racial discrimination in a healthcare setting. This is a relevant and a not well-studied topic and the findings of the paper should be taking account for the future. However, there is a lack of some relevant factors that must be detail and explain.

We appreciate your thorough review. We address each of your critiques below, and in the process have substantially strengthened our manuscript.

There is some typos/ deleted references f.e MMT vs. no MMT [ref[), see all the mns.

Our use of “[ref]” in the description of categorical variables refers to the referent group chosen for analyses of categorical variables. We have spelled out the word ‘referent’ in place of ‘ref’ so as not to be confused with a placeholder for a missing literature reference. (Lines 161-168)

The expression “mental health disorder diagnosis” should be change by Dual Disorders, in the text, tables, etc for a review of the use of this term see Szerman et al. Rethinking Dual disorders/pathology. Addictive Disorders & Their Treatment. 2013; 12 (1): 1–10 doi: 10.1097/ADT.0b013e31826e7b6a

Thank you for helping us use the correct terminology of dual disorders. We have replaced ‘mental health disorder diagnosis’ with ‘dual disorders’ throughout the manuscript text and tables.

It is necessary to use the term Latino/Latina? Or Latino includes all?

There are several ways in which social scientists label the Latino/Latina group in the U.S. We opted for Latino/Latina as it includes both male (Latino) and female (Latina) versions of this label, and generally refers to people descended from populations in Latin America. Other common labels include Hispanic (people who speak Spanish and come from Spanish-speaking populations) and Latinx (a non-gendered version of Latino/Latina). 

Introduction

Reviewer 1

The literature discussed is entirely North American. This should be acknowledged in the text; differences in patterns of use and misuse vary world-wide.

We have reiterated throughout the text that both our literature review and data are U.S.-based. We also added an indication of our U.S. focus in the manuscript title. 

The term ‘drug abuse’ is not generally accepted in other regions. ‘Misuse’ is considered more acceptable.

Thank you for pointing out these important differences. We have replaced ‘abuse’ with ‘misuse’, except in circumstances where we are referencing the work of others who used the term ‘abuse’ in their own work. (Line 49)

The word ‘prevalence’ should be used in place of ‘incidence’ for heroin and opioid misuse, as the paper referenced does not measure incidence.

We have replaced ‘incidence’ with ‘prevalence’ in our references to both the Jones et al. (2015) and Krawczyk et al. (2017) papers. (Lines 49 and 81)

Similarly, poly substance use is a risk for overdose, rather than opioid misuse; the factors listed are linked to prevalence of misuse, rather than attributed to those factors, s the causal linkages may be bidirectional for many of them.

Thank you for clarifying the inaccuracies in this sentence. We have changed ‘attributed’ to ‘linked’, and restructured this sentence to more accurately reflect the relationships between factors linked to misuse. (Lines 51-55)

Paragraph 2 starts talking about rates of OD deaths, but gives an example of the rate of change in the number of OD deaths. These concepts are related but not interchangeable.

In support of our statement that overdose death rates are higher among non-Whites in some states, we have added a Kaiser Family Foundation citation to illustrate state-level racial differences in the rate of overdose deaths. We also clarified the use of rate versus rate of change. (Lines 59-63)

Similarly in paragraph 3, the authors switch between discussions of MMT (specifically methadone) and other MATs (medication assisted treatments, which include other OSTs).

Thank you for pointing out the difference between MMT and MAT. We have made clear that MMT is the focus of this study, but is also only one type of MAT. The NESARC-III dataset only includes data on methadone-specific maintenance therapy. We further justified our use of citations that include different types of MAT. In the context of this MMT study, better understanding disparities in both MMT and other MAT types is useful in the overall narrative of OUD treatment disparities and discrimination in healthcare settings. (Lines 68-72)

In paragraph 4, “experiences of racial discrimination ….more common… racial/ethnic minorities”… but a comparison group is not cited.

We have added that the comparison being made is between racial/ethnic minority groups and Whites. (Line 85)

McNight’s paper apparently found that discrimination differed between groups, but which groups?

We have included additional findings from the McKnight paper, specifically that Latino/Latina drug users were the group least likely to perceive institutional racism in healthcare settings. (Lines 89-90)

In paragraph 5 is it possible the text should read “differences in the effects on MMT of discrimination? The following sentence/s refers to the link between race and receipt of appropriate MMT.

Thank you for pointing out the confusing structure of this sentence and paragraph. American Indian/Alaska Natives are excluded from OUD disparities research in general, and MMT research in particular, often due to small sample sizes or unavailable data for this group. We have clarified what we believe to be the important takeaway message in this paragraph – although there are few MMT studies that compare AI/AN to other groups, we do see some reports that some AI/AN MMT clients have reported institutional discrimination in the extant literature. (Lines 102-104)

In the final paragraph of the Background, it is somewhat tautological that racial/ethnic minority status would be associated with racial discrimination.

Thank you for pinpointing the ambiguous nature of this part of our hypothesis. We added language to clarify our hypothesized association between race/ethnicity and discrimination – i.e., compared to Whites, we hypothesized that racial/ethnic minority groups would have higher odds of racial discrimination in healthcare settings, and the odds ratio estimates would vary in magnitude by racial group. (Lines 115-118)

Reviewer 2

The authors could include some of the following references as part of the citations of the effects on the positive effects of medication-assisted treatment.

Ma J, Bao YP, Wang RJ, et al. Effects of medication-assisted treatment on mortality among opioids users: a systematic review and meta-analysis. Molecular psychiatry. 2018. Maglione MA, Raaen L, Chen C, et al. Effects of medication assisted treatment (MAT) for opioid use disorder on functional outcomes: A systematic review. Journal of substance abuse treatment. 2018;89:28-51.

Thank you for bringing these references to our attention. We have added a reference to the positive effect of MAT on functional outcomes (Maglione et al., 2018). (Line 77) 

The meta-analysis by Ma et al. (2018) indeed demonstrates a positive effect of MAT on reduced mortality. However, only four out of 30 studies included in their meta-analysis were based in the United States. We previously responded to Reviewer 1’s concern (Comment #1) that our data source is entirely based in North America, and that disparities in OUD and MAT utilization vary worldwide. Our response to Reviewer 1 was that we have made it clear throughout our manuscript that our literature review, data, and discussion are all in the context of North America. In this sense, a reference to Ma et al. may be out of place within our focus on MAT in the US.

Reviewer 3

Read: McKnight et al. Perceived discrimination among racial and ethnic minority drug users and the association with health care utilization. J Ethn Subst Abuse. 2017;16(4):404-419. doi: 10.1080/15332640.2017.1292418.

Thank you for reiterating the importance of the McKnight et al. (2017) article. In response to a comment by Reviewer 1 above (Comment #8), we have added to our introduction the specific findings of racial/ethnic differences in perceived institutional racial discrimination. Specifically, Latino/Latina drug users were the least likely to perceive institutional racism in health service settings. (Lines 89-90)

Methods

Reviewer 1

Variables in NESARC describe substance use disorders etc, rather than ‘addressing’ them.

We have changed ‘addressing’ to ‘describing’. (Line 125)

Stratum is a singular noun – perhaps should be strata.

Indeed, in this sentence we intended to use a plural noun. We have changed ‘stratum’ to ‘strata’. (Line 127)

In line 136, perhaps use gender rather than ‘sex’, to avoid confusion with ‘sexual minority status’, which is presumed to refer to sexual orientation.

While we do our best to recognize the differences in the application of ‘sex’ versus ‘gender’, the NESARC-III questionnaire asks respondents to report their sex, including only two response options of male or female. Therefore, we have opted to continue using ‘sex’ for this study, instead of ‘gender’, in order to more accurately reflect the survey questionnaire. 

We also clarified the construction of our sexual minority status variable, so as to avoid any confusion with other covariates. Survey respondents were asked about their sexual orientation, with response options including heterosexual or straight [referent], or gay/lesbian, bisexual, or not sure. We categorized respondents who identified as non-heterosexual as sexual orientation minorities.

In the Analysis section, why was a cut-off of 0.2 used for the p-values for inclusion in the model, rather than the usual 0.05?

While there are many useful strategies for variable selection in multivariable regression models, we followed the strategy of ‘purposeful selection of covariates’ outlined in the textbook Applied Logistic Regression by David Hosmer and Stanley Lemeshow (2013). In using this approach, we recognize that a cutoff of the traditional α=0.05 level may fail to identify variables known to be important, and that may demonstrate a significant degree of confounding in the presence of other covariates. At the same time, a threshold of α=0.20 is low enough as to avoid overfitting a model. We have included a description of this approach in our methods section, as well as a reference for the source material. (Lines 175-177)

Reviewer 2

It would be important to explain / operationalize racial discrimination in healthcare settings- do these include discrimination in all types of health care settings including substance abuse specialty treatment centers, that is centers where methadone might be dispense or does it exclude it? A clear explanation of what healthcare settings includes can help in interpreting the findings and understanding their full implication.

The two NESARC-III survey questions that ask about racial discrimination in healthcare settings do not specify the specific type of healthcare setting. We have added a statement in this paragraph that the type of healthcare setting was not specified in these survey questions, leaving the interpretation of what constitutes a healthcare setting up to the survey respondent. Additionally, we restructured our description of the contents of these survey questions for clarity. (Lines 140-149)

Reviewer 3

Page 6 “Our outcome of interest was a binary indicator of whether an individual experienced 120 racial discrimination in a healthcare setting in the past year (yes or no). Response options to 121 discrimination survey questions included very often, fairly often, sometimes, almost never, and 122 never. Our binary variable included those who endorsed no experience (never) or any experience…”

I suppose that that no racial discriminations means that the answer is “never” and yes means “very often, fairly often, sometimes, almost never”, in any of the questions, please clarify this sentence.

Thank you for helping us clarify how we constructed our discrimination variable. We have added that our ‘any’ discrimination variable level includes very often, fairly often, sometimes, or almost never. Also, please see our response to Reviewer 2 (Comment #17), which also addresses strategies to more clearly articulate the construction of this variable. 

Could you show if there are differences between patients on MMT coming from heroin-dependence or opiate pain-killer dependence?

We appreciate your inquiry into the differences between heroin and opioid painkiller dependence, as our current sample is restricted to anyone who had a lifetime heroin or other opioid use disorder.

In recognition of the clinical and behavioral differences between the two drug types, we opted to include an additional covariate in our analysis, indicating whether an individual had a heroin use disorder only, an opioid use disorder only, or both. We added the prevalence and chi-square estimates to Table 1 (Line 191). We found that the specific opioid use disorder type was not significantly associated with past-year experiences of racial discrimination in healthcare settings (x2 = 0.26, p = 0.8801). This variable did not meet our selection criteria for inclusion in our fitted model, leaving our fitted model unchanged.

Results

Reviewer 1

Does Table 2 represent a single multivariable model, adjusted for all listed variables and covariates? If so, why does line 200 in the Discussion refer to ‘models’?

Thank you for catching this typo. Table 2 represents a single multivariable model with an interaction term. We have changed ‘models’ to ‘model’. (Line 230)

Discussion

Reviewer 1

In general, a note that the relationships explored in this analysis are associations, and causality cannot be assumed. Therefor (line 201) one variable should not be discussed as having an effect on another.

Thank you for helping us correct this mistake. We have changed ‘effect of’ to ‘association between’. (Line 228) 

Line 208: the Fitzgerald article describes occasions of bias, rather than trends (changes).

We have changed ‘trends in implicit biases’ to ‘occasions of implicit bias’. (Line 239)

Line 210: …”people with marginalized characteristics” should read "marginalizing characteristics"

Thank you. We have changed ‘marginalized’ to ‘marginalizing’. (Line 242)

Paragraph 3 of the Discussion draws some long bows without references, e.g. the experience of discrimination negatively affecting compliance (there are references for this) and the “worsening of MMT comorbidities” (unspecified) which may refer to comorbid conditions going untreated if people avoid healthcare in general after experiencing discrimination.

We have added two references in support of our claim that discrimination negatively effects compliance, including Lee et al. (2009) and Burgess et al. (2008). Including these references more effectively grounds our findings in the extant literature. We also added to this point by suggesting that treatment addressing MMT comorbidities may be delayed or foregone as a result of prior experiences with discrimination. (Lines 254-256)

Line 223 may read more clearly as “..the convergence in AI/AN communities of the opioid epidemic, multiple barriers… “ etc as the ‘opioid epidemic’ is not specific to those communities.

Thank you for helping to clarify this sentence. We have changed ‘the convergence of the opioid epidemic in AI/AN communities…’ to ‘the convergence in AI/AN communities of the opioid epidemic…’. (Line 258)

The final statements in this paragraph seem to suggest that treatment of conditions comorbid with OUDs must precede treatment for the OUDs: this is an outdated concept. There is considerable literature suggesting that it is more effective to address multiple co-occurring conditions in parallel or in integrated fashion, rather than sequentially.

We have removed the outdated citations and replaced them with references that support the modern way practitioners think about OUD treatment and integrated behavioral health settings. The Abraham et al. (2017) article discusses the importance of integrated care for the treatment of OUD, as well as how the Affordable Care Act has been key in moving treatment services forward towards more integrated service delivery models. We also included a citation by Chou et al. (2016), which outlines strategies for improved care of people with OUDs in integrated primary care settings. (Lines 260-266)

The final comments relate to challenges faced in service delivery for AI/AN communities. There is something of a leap in stating that the findings from this study justify initiatives targeting Native American populations, however essential those initiatives are.

We agree that promoting disease prevention initiatives among AI/AN is important, but concluding that our study justifies the existence of those initiatives is indeed beyond the scope of our findings. Rather than remove this section entirely, we thought that an alternative framing around AI/AN populations would serve as an appropriate end to the discussion section. Instead of stating that our findings justify such AI/AN-focused programs, we added that our findings may inform future research into specific programs addressing MAT, health services, and improved health within AI/AN communities. (Lines 279-285)

Reviewer 2

A stronger justification towards the findings in AI/AN is warranted. The statement “our strong findings of racial/discrimination in healthcare settings among AI/AN may be driven, in part, by AI/AN biases inherent in non-IHS services” requires a citation and a broader explanation to contextualize the findings.

We have added two articles as references to support our statement that our AI/AN findings may be driven by biases in non-IHS settings (Bailey et al., 2017; Walls et al., 2015). Both articles demonstrate racial and institutional baises experienced by AI/AN. Bailey et al. looked at health equities in the US in general but highlighted inequities particularly relevant to AI/AN. Walls et al. looked at racial microaggressions experienced by AI/AN, and demonstrated correlations between microaggressive/discriminatory behaviors towards AI/AN and poor health outcomes. We have also added some additional explanation to contextualize these references, including an examination of IHS services coverage among AI/AN and the potential for a high risk of experiencing discriminatory behavior in non-IHS services. (Lines 277-281)

Reviewer 3

I think that a short sentence summarizing all your relevant results it will do the reading more clear. After that you can discuss your results.

Thank you for helping us begin our discussion section with a succinct description of our main finding. We have reorganized this first paragraph to begin with summarizing our main finding, then delving into specific interaction findings with a focus on our strong AI/AN results. (Lines 223-233)

Your results “Experiences of racial discrimination were more likely among …those with a past-year mental health) diagnosis (p<0.0001)”. I believe that is an important factor and any comments should be added on this topic.

Do you any date or any comments if there is an association between dual disorders / raze-ethnicity and discrimination??

Could you speculate if experiences of racial discrimination were more likely among any raze of dual disorders opiate-dependent? Should be done and special intervention or screening on discrimination in this patients?

Thank you for pointing out the high prevalence of past-year dual mental health diagnoses among OUD treatment participants. In our fitted model and among our covariates, the odds of experiences of racial discrimination were highest among those with dual mental health diagnoses (versus none) compared to the estimates of all other covariates. Clearly, dual diagnoses play an integral role in MMT services utilization and healthcare-based discrimination. We have added a line in the discussion that reiterates the importance of our descriptive finding. (Lines 251-252)

Yes, it is possible to look at differences in the association between discrimination and dual disorder status, dependent on race/ethnicity. One approach to solve this problem would be adding a second interaction term to our fitted model, which would be race/ethnicity by dual diagnosis status (similar in structure to our original race/ethnicity by MMT interaction). However, while this is an important research question addressing a gap in the MMT disparities literature, we believe that the inclusion of a second interaction term would distract from the stated scope of this paper – i.e., differences in associations between MMT and discrimination within racial/ethnic groups. Instead, we plan on pursuing a future manuscript that will look at disparities in MMT utilization based on dual disorder status. 

Limitations

Reviewer 1

The Limitations section should include a statement noting that relationships cannot be construed as causal, and that data are from the USA only and may not generalise to other settings.

Thank you for pointing out some of the inherent limitations of this North American-based, cross-sectional dataset. We have added a paragraph articulating the limits of causal inference, as well as our study’s limited generalizability. (Lines 297-300)

Reviewer 2

Authors should acknowledge that the dataset ( NESARC- III) is from 2012-2013, and while racial discrimination in healthcare settings by individuals on MMT might be pervasive and not easy to change, multiple national initiatives towards the opioid epidemic and education towards evidence based treatments within mainstream of care might shift the results observed, should we look at current data.

We have included a section that points out the possibility of different findings if a similar or replicated study were based on more current data. (Lines 300-304)

Conclusion

Reviewer 1

The Conclusion refers to issues that are worthy of attention but extrapolates somewhat from the findings, as it appears to infer that it is racial discrimination that is the major barrier to health service uptake, particularly OST, rather than this as perhaps an expression of the multiple other stigma associated with illicit drug use.

Thank you for your insight into making our conclusion stronger. We have restructured this section to better reflect the findings and implications of our study. In particular, we noted the challenges of interpreting one expression of stigma (racial discrimination) in the midst of other unobserved or latent factors that may also affect how racial discrimination is both expressed and perceived in healthcare settings. (Lines 311-323)

Finally - some attention should be paid to accuracy and consistency of formatting of the references.

We have reviewed and updated our reference list for accuracy and consistency, and to the best of our knowledge all of our references follow the PLOS ONE guidelines, specifically using the Vancouver citation style.

Also, in response to Reviewer 3’s general feedback, we have changed the way we refer to reference groups for categorical variables in the methods section, so as to avoid being mistaken as a placeholder for a missing citation.

---

## [Decision Letter · Decision Letter 1]

9 Dec 2019

PONE-D-19-17408R1

Interaction effects in the association between methadone maintenance therapy and experiences of racial discrimination in U.S. healthcare settings

PLOS ONE

Dear Dr. Pro,

Thank you for submitting your manuscript to PLOS ONE. After careful consideration, we feel that it has merit but does not fully meet PLOS ONE’s publication criteria as it currently stands. Therefore, we invite you to submit a revised version of the manuscript that addresses the points raised during the review process.

We would appreciate receiving your revised manuscript by Jan 23 2020 11:59PM. To enhance the reproducibility of your results, we recommend that if applicable you deposit your laboratory protocols in protocols.io, where a protocol can be assigned its own identifier (DOI) such that it can be cited independently in the future. For instructions see: http://journals.plos.org/plosone/s/submission-guidelines#loc-laboratory-protocols

We look forward to receiving your revised manuscript.

Kind regards,

Stéphanie Baggio

Academic Editor

PLOS ONE

Additional Editor Comments (if provided):

Unfortunately, two of the previous reviewers were unable to assess the manuscript again. I assessed it myself and I also added some additional (minor) comments.

1. Please report p<.001 (three decimals) in the abstract, results, and tables.

2. Even if the study’s methods have been published elsewhere, the response rate should be mentioned.

3. Which kind of survey weights did you use?

4. More information is needed for the measures: a reference for the racial discrimination scale and which kind of diagnosis (DSM-5?) for addictive disorders. This is especially important because the definition changed between DSM-IV and DSM-5.

Reviewers' comments:

Reviewer's Responses to Questions

**Comments to the Author**

1. If the authors have adequately addressed your comments raised in a previous round of review and you feel that this manuscript is now acceptable for publication, you may indicate that here to bypass the “Comments to the Author” section, enter your conflict of interest statement in the “Confidential to Editor” section, and submit your "Accept" recommendation.

Reviewer #3: (No Response)

2. Is the manuscript technically sound, and do the data support the conclusions?

Reviewer #3: Yes

3. Has the statistical analysis been performed appropriately and rigorously? 

Reviewer #3: Yes

4. Have the authors made all data underlying the findings in their manuscript fully available?

Reviewer #3: No

5. Is the manuscript presented in an intelligible fashion and written in standard English?

Reviewer #3: Yes

6. Review Comments to the Author

Reviewer #3: The paper could be accepted but yhe authors dosen´t respond all my concerns. However, a moderate review must be done and after that should be reviewed again.

This paper provides some evidence on the role of race and ethnicity on discrimination among opiate-dependent patients in MMT. Twenty-two percent of the sample experienced racial discrimination in a healthcare setting. This is a relevant an a not well-studied topic and the findings of the paper should be taking account for the future. However, there is a lack of some relevant factors that must be detail and explain. The authors confirm my guess that the dual-diagnosis patients have more discrimination. Authors doesn´t replay all my concerns and comments so I think that the paper should be reviewed again.

General comments

Abstract

It is clear according the result and discussion, but should be modified if there is new comments or results.

Introduction

That´s ok

Methods & Results:

Could you show if there are differences between patients on MMT coming from heroin-dependence or opiate pain-killer dependence?

“We appreciate your inquiry into the differences between heroin and opioid painkiller

dependence, as our current sample is restricted to anyone who had a lifetime heroin or

other opioid use disorder.”

The pain killers are opioids so I guess that you want to said.

“We appreciate your inquiry into the differences between heroin and opioid painkiller

dependence, as our current sample is restricted to anyone who had a lifetime heroin or

other (illegal) opioid use disorder.” Please if it Ii right change explain it in the paper.

Discussion

The first short sentence summarizing all your relevant results it is very clear.

The presence of Dual disorders is high as expected is close to 60% last year, according other research among opiate-dependent patients see Roncero et al. Psychiatric comorbidities in opioid-dependent patients undergoing a replacement therapy programme in Spain: The PROTEUS study. Psychiatry Res. 2016 Sep 30;243:174-81. doi:10.1016/j.psychres.2016.06.024. and Martínez-Luna et al. Harm reduction program use, psychopathology and medical severity in patients with methadone maintenance treatment. Adicciones. 2018 Jan 15;30(3):197-207. doi: 10.20882/adicciones.897.

I agree with the new sentence, but, should be point out that confirm previous findings, r please add the references and the idea that it is expected

For example, results from our descriptive analyses indicated that, as expected (references..) , nearly 60% of OUD treatment participants also had a dual mental health diagnosis in the past year.

Your results “Experiences of racial discrimination were more likely among …those with a past-year mental health) diagnosis (p<0.0001)”. I believe that is an important factor an any comments should be added on this topic. Do you any date or any comments if there is an association between dual disorders / raze-ethnicity and discrimination?? Could you speculate if experiences of racial discrimination were more likely among any raze of dual disorders opiate-dependent? Should be done an special intervention or screening on discrimination in this patients?

Your Response “… it is possible to look at differences in the association between discrimination and dual disorder status, dependent on race/ethnicity. One approach to solve this problem

would be adding a second interaction term to our fitted model, which would be race/ethnicity by dual diagnosis status (similar in structure to our original race/ethnicity by MMT interaction). However, while this is an important research question addressing a gap in the MMT disparities literature, we believe that the inclusion of a second interaction term would distract from the stated scope of this paper – i.e., differences in associations between MMT and discrimination within racial/ethnic groups. Instead, we plan on pursuing a future manuscript that will look at disparities in MMT utilization based on dual disorder status. “

I understand that due to the space it is not possible to show here the dates that you have. I encourage you to do the second paper¡¡, because the dual-disorders patients for sure are suffering the double-discrimination and double-stigma and it is important to show it to the scientific and clinicians community. But at least in the paper it important to describe the relevance of this topic and in the last paragraph to add something as in the future the interaction between the racial and dual-disorders diseases should be studied… or something like that.

You should analyze not only the dual-disorders, if it is possible ( according the “n” of the sample, psychotic versus affective disorders vs personality disorders… obviously depending of the prevalence in your sample

It seems that there is no discrimination according sex, it is important to point out this result.

Conclusions:

Take in account my previous suggestions

7. PLOS authors have the option to publish the peer review history of their article (what does this mean?). If published, this will include your full peer review and any attached files.

Reviewer #3: No

---

## [Author Response · Author response to Decision Letter 1]

3 Jan 2020

METHODS

1. Even if the study’s methods have been published elsewhere, the response rate should be mentioned.

We have included a sentence describing the household survey response rate of 72%.

2. Which kind of survey weights did you use?

We have clarified that we used survey weights pre-calculated by NESARC which are designed to calibrate the sampling distribution for major subgroups defined by region, sex, age, and race/ethnicity. 

3. More information is needed for the measures: a reference for the racial discrimination scale and which kind of diagnosis (DSM-5?) for addictive disorders. This is especially important because the definition changed between DSM-IV and DSM-5.

We have clarified that the reference group for our binary discrimination indicator is ‘no’. We have also included additional text describing the pre-defined NESARC diagnostic variables based on DMS-5 criteria.

4. Could you show if there are differences between patients on MMT coming from heroin-dependence or opiate pain-killer dependence?

Our sample is restricted to only those who meet the criteria for heroin use disorder or illicit opioid use disorder, based on DSM-5 criteria. Our analysis includes a covariate differentiating between opioid use disorder, illicit opioid use disorder, or both. In our bivariate analyses (Table 1), we found no association between disorder type and discrimination in a healthcare setting (x2=2.45, p=0.458). Thus, this variable was not carried forward to our fitted model. While the association between disorder type and discrimination may vary by MMT status, as you suggest, we had no theoretical or algorithmic justification to explore this relationship as disorder type demonstrated a very weak relationship with our outcome.

RESULTS

5. Please report p<.001 (three decimals) in the abstract, results, and tables.

We have changed the decimal place of our p-values to only include three decimal places.

DISCUSSION

6. The presence of Dual disorders is high as expected is close to 60% last year, according other research among opiate-dependent patients see Roncero et al. Psychiatric comorbidities in opioid-dependent patients undergoing a replacement therapy programme in Spain: The PROTEUS study. Psychiatry Res. 2016 Sep 30;243:174-81. doi:10.1016/j.psychres.2016.06.024. and Martínez-Luna et al. Harm reduction program use, psychopathology and medical severity in patients with methadone maintenance treatment. Adicciones. 2018 Jan 15;30(3):197-207. doi: 10.20882/adicciones.897. You should point out that confirm previous findings, r please add the references and the idea that it is expected.

Thank you for helping us provide more context around our strong finding of dual mental health disorders and experiences of discrimination. We have cited the articles by Roncero, et al. and Martinez-Luna, et al. 

7. Your results “Experiences of racial discrimination were more likely among those with a past-year mental health diagnosis (p<0.0001)” are an important factor and any comments should be added on this topic. Do you any date or any comments if there is an association between dual disorders / raze-ethnicity and discrimination? I understand that due to the space it is not possible to show here the dates that you have. I encourage you to do the second paper!!

Thank you for encouraging us to pursue this topic further. A second paper addressing dual mental health diagnoses is forthcoming. We have added comments and citations to highlight the importance of mental health comorbidities in among our target population.

8. It is important to describe the relevance of dual mental health disorders, and in the last paragraph to add something as in the future the interaction between the racial and dual-disorders diseases should be studied… or something like that.

We agree that this is an important topic for future research. We have added text outlining a possible future research trajectory which could be designed to answer questions about the relationships between race/ethnicity, mental health, stigma, discrimination, and MMT services utilization. 

9. It seems that there is no discrimination according sex, it is important to point out this result.

In our fitted model, we found that women have lower odds of experiencing racial discrimination, compared to men (aOR=0.51, p<0.001). We have added a paragraph highlighting this finding, and provided additional citations to contextualize these results within the larger body of extant literature addressing differential MMT outcomes by gender.

---

## [Decision Letter · Decision Letter 2]

23 Jan 2020

Interaction effects in the association between methadone maintenance therapy and experiences of racial discrimination in U.S. healthcare settings

PONE-D-19-17408R2

Dear Dr. Pro,

We are pleased to inform you that your manuscript has been judged scientifically suitable for publication and will be formally accepted for publication once it complies with all outstanding technical requirements.

With kind regards,

Stéphanie Baggio

Academic Editor

PLOS ONE

Additional Editor Comments (optional):

Reviewers' comments:

Reviewer's Responses to Questions

**Comments to the Author**

1. If the authors have adequately addressed your comments raised in a previous round of review and you feel that this manuscript is now acceptable for publication, you may indicate that here to bypass the “Comments to the Author” section, enter your conflict of interest statement in the “Confidential to Editor” section, and submit your "Accept" recommendation.

Reviewer #3: All comments have been addressed

2. Is the manuscript technically sound, and do the data support the conclusions?

Reviewer #3: Yes

3. Has the statistical analysis been performed appropriately and rigorously? 

Reviewer #3: Yes

4. Have the authors made all data underlying the findings in their manuscript fully available?

Reviewer #3: Yes

5. Is the manuscript presented in an intelligible fashion and written in standard English?

Reviewer #3: Yes

6. Review Comments to the Author

Reviewer #3: (No Response)

7. PLOS authors have the option to publish the peer review history of their article (what does this mean?). If published, this will include your full peer review and any attached files.

Reviewer #3: No

---

## [Editor Report · Acceptance letter]

30 Jan 2020

PONE-D-19-17408R2 

Interaction effects in the association between methadone maintenance therapy and experiences of racial discrimination in U.S. healthcare settings 

Dear Dr. Pro:

I am pleased to inform you that your manuscript has been deemed suitable for publication in PLOS ONE. Congratulations! Your manuscript is now with our production department. 

With kind regards,

on behalf of

Dr. Stéphanie Baggio 

Academic Editor

PLOS ONE